# Propositional Kernels

**DOI:** 10.3390/e23081020

**Published:** 2021-08-07

**Authors:** Mirko Polato, Fabio Aiolli

**Affiliations:** Department of Mathematics, University of Padova, 35143 Padova, Italy; aiolli@math.unipd.it

**Keywords:** propositional kernels, boolean kernels, kernel methods, categorical data, propositional logic

## Abstract

The pervasive presence of artificial intelligence (AI) in our everyday life has nourished the pursuit of explainable AI. Since the dawn of AI, logic has been widely used to express, in a human-friendly fashion, the internal process that led an (intelligent) system to deliver a specific output. In this paper, we take a step forward in this direction by introducing a novel family of kernels, called Propositional kernels, that construct feature spaces that are easy to interpret. Specifically, Propositional Kernel functions compute the similarity between two binary vectors in a feature space composed of logical propositions of a fixed form. The Propositional kernel framework improves upon the recent Boolean kernel framework by providing more expressive kernels. In addition to the theoretical definitions, we also provide an algorithm (and the source code) to efficiently construct any propositional kernel. An extensive empirical evaluation shows the effectiveness of Propositional kernels on several artificial and benchmark categorical data sets.

## 1. Introduction

Explainable Artificial Intelligence has become a hot topic in the current research [1,2]. The recent pervasive deployment of machine learning solutions has hugely increased the need of having transparent and easy-to-explain models. Many efforts have been devoted to improving the interpretability of complex and non-linear models like (deep) neural networks [3]. Despite the effort, we are still far from being able to systematically interpret what happens under the hood. Still, there are specific applications in which explainability is not only a desideratum but a necessity. For instance, in medical applications, physicians need specific and comprehensible reasons to accept the prediction given by a machine. For this reason, models like Decision Trees (DT) are widely used by non-expert users thanks to their easy logical interpretation. The shortcoming of DTs is that, in general, they are inferior models w.r.t. neural networks or Support Vector Machines (SVM).

Neural networks are not the only (non-linear) models that are hard to interpret. Kernel machines, like SVMs [4,5,6], typically work on an implicitly defined feature space by resorting to the well-known kernel trick. The use of such an implicit representation clearly harms the interpretability of the resulting model. Moreover, these feature spaces are often high dimensional and thus this makes the job even harder. Nonetheless, in the last decade, several methods have been introduced for extracting rules from SVMs [7]. Typically, these techniques try to build if-then-else rules over the input variables, but the task is not trivial since the feature space might be not easy to deal with in terms of explanatory capabilities.

Given binary-valued input data, a possible approach to make SVM more interpretable consists of defining features that are easy to interpret, for example, features that are logical rules over the input. To this end, recently, a novel Boolean kernel (BK) framework has been proposed [8]. This framework is theoretically well-founded and the authors also provide efficient methods to compute this type of kernels. Boolean kernels are kernel functions in which the input vectors are mapped into an embedding space formed by logical rules over the input variables, and, in such space, the dot product is performed. In other words, these kernels compute the number of logical rules of a fixed form over the input variables that are satisfied in both the input vectors. To show that Boolean kernels can hugely improve the interpretability of SVM, in [9] a proof of concept method based on a genetic algorithm has been proposed. This algorithm can extract from the hypothesis of an SVM the most influential features (of the feature space) in the decision. Being those features logical rules over the input, they can be used to explain the SVM’s decision.

However, the main drawback of Boolean kernels is the limited logical expressiveness. Albeit the Boolean kernel framework allows defining Disjunctive Normal Form (DNF) kernels, only a subset of all the possible DNF can be formed. This limitation is due to the way Boolean kernels are computed and in Section 3.1 we show that such a limitation cannot be overcome using the BK framework itself.

For this reason, in this paper, we improve upon BKs by introducing a new family of kernels, called Propositional Kernels [10]. Differently from BKs, with Propositional Kernels is possible to define kernels for any propositional logic formula (formulas with bound variables are included in the feature space but they are not exclusive). Starting from the main limitations of the BK framework, we show step-by-step how propositional kernels are computed. Then, we provide a mathematical definition for all possible binary logical operations as well as a general procedure to compose base Propositional kernels to form any propositional kernel. This “composability” makes Propositional kernels highly expressive, much more than BKs.

In the experimental section, we evaluate the effectiveness of these new kernels on several artificial and benchmark categorical data sets. Moreover, we propose a heuristic, based on a theoretically grounded criterion, to select good propositional kernels given the data set. We empirically show that this criterion is usually able to select, among a set of randomly generated Propositional kernels, the one that performs best. This framework is open-source, and it is publicly available at https://github.com/makgyver/propositional_kernels (accessed on 6 August 2021).

The rest of the paper is structured as follows. In Section 2 we review the related work, while in Section 3 we provide an overview of the Boolean kernel framework giving a particular emphasis to the limitations of such framework. Then, Section 4 presents the main ideas behind Propositional kernels and how to compute all kernels corresponding to binary and unary logical operations. Section 5 demonstrates how to compose base Propositional kernels to obtain any possible Propositional kernel. The empirical evaluation is presented in Section 6, and finally, Section 7 wraps up the paper and provides some possible future directions.

## 2. Related Work

The concept of Boolean kernel, i.e., kernels with a feature space representing Boolean formulas over the input variables, has been introduced in early 2000 by Ken Sadohara [11]. He proposed an SVM for learning Boolean functions: since every Boolean (i.e., logic) function can be expressed in terms of Disjunctive Normal Form (DNF) formulas, the proposed kernel creates a feature space containing all possible conjunctions of negated or non-negated Boolean variables.

For instance, given $x\in R^{2}$, the feature space of a DNF kernel with a single conjunction contains the following $3^{2}-1$ features: $x_{1},x_{2},\neg x_{1},\neg x_{2},x_{1}\wedge x_{2},x_{1}\wedge \neg x_{2},\neg x_{1}\wedge x_{2},\neg x_{1}\wedge \neg x_{2}$.

Formally, the DNF and the monotone DNF (mDNF) kernel, i.e., DNF with no negated variables, between $x,z\in R^{n}$ are defined as
$$\kappa_{\mathrm{DNF}}\left(x,z\right)=-1+\prod_{i=1}^{n}\left(2x_{i}z_{i}-x_{i}-z_{i}+2\right),\;\kappa_{\mathrm{mDNF}}\left(x,z\right)=-1+\prod_{i=1}^{n}\left(x_{i}z_{i}+1\right).$$

Sadohara’s DNF kernel works on real vectors, thus conjunctions are multiplications and disjunctions are summations. Using such a DNF kernel, the resulting decision function of a kernel machine can be represented as a weighted linear sum of conjunctions (Representer Theorem [12,13]), which in turn can be interpreted as a “soft” DNF. The DNF kernel, restricted to binary inputs, has been independently discovered in [14,15]. Its formulation is a simplified version of the Sadohara’s DNF kernel:$$\kappa_{\mathrm{DNF}}^{\prime }\left(x,z\right)=-1+2^{\langle x,z\rangle +\langle 1-x,1-z\rangle },\;\kappa_{\mathrm{mDNF}}^{\prime }\left(x,z\right)=-1+2^{\langle x,z\rangle }.$$

A drawback of these types of kernels is the exponential growth of the size of the feature space w.r.t the number of involved variables, i.e., $3^{n}-1$ for *n* variables. Thus, the similarity between the two examples is equally influenced by simple DNFs over the input variables, as well as very complex DNFs. To give the possibility of controlling the size of the feature space, i.e., the involved DNFs, Sadohara et al. [16] proposed a variation of the DNF kernel in which only conjunctions with up to *d* variables (i.e., *d*-ary conjunctions) are considered. Over binary vectors, this kernel, dubbed d-DNF kernel, is defined as
κDNFd(x,z)=∑i=1d〈x,z〉+〈1−x,1−z〉i,

Following the same idea of Sadohara, Zhang et al. [17] proposed a parametric version of the DNF kernel for controlling the influence of the involved conjunctions. Specifically, given $x,z\in {\left\{0,1\right\}}^{n}$ and σ>0, then
$$\begin{array}{cc} \kappa_{\mathrm{DNF}}^{\left(\sigma \right)}\left(x,z\right) & =-1+\prod_{i=1}^{n}\left(\sigma x_{i}z_{i}+\sigma \left(1-x_{i}\right)+\sigma \left(1-z_{i}\right)+1\right), \end{array}$$
where σ induces an inductive bias towards simpler or more complex DNF formulas.

An important observation is that the embedding space of a classical non homogeneous polynomial kernel of degree *p* is composed of all the monomials (i.e., conjunctions) up to the degree *p*. Thus, the only difference between the polynomial and the d-DNF kernel is the weights associated to the features.

A kernel closely related to the polynomial kernel is the all-subset kernel [18,19], defined as
$$\kappa_{\subseteq }\left(x,z\right)=\prod_{i=1}^{n}\left(x_{i}z_{i}+1\right).$$

This kernel has a feature space composed of all possible subsets of the input variables, including the empty set. It is different from the polynomial kernel because it does not limit the number of considered monomials/subsets, and all features are equally weighted. We can observe that the all-subset kernel and the monotone DNF kernel are actually the same kernel up to the constant −1, i.e., $\kappa_{\subseteq }\left(x,z\right)=\kappa_{\mathrm{mDNF}}\left(x,z\right)+1$. A common issue of both the polynomial and the all-subsets kernel is that they have limited control of which features they use and how they are weighted.

A well-known variant of the all-subset kernel is the ANOVA kernel [18] in which the feature space is formed by monomials of a fixed degree without repetitions. For instance, given $x\in R^{3}$ the feature space induced by the all-subset kernel would have the features $x_{1},x_{2},x_{3},x_{1}x_{2},x_{1}x_{3},x_{2}x_{3},x_{1}x_{2}x_{3}$ and *∅*, while the feature space of the ANOVA kernel of degree 2 it would be made up by $x_{1}x_{2},x_{1}x_{3}$ and $x_{2}x_{3}$.

Boolean kernels have also been used for studying the learnability of logical formulae using maximum margin algorithms, such as SVM [20,21]. Specifically, [21] shows the learning limitations of some Boolean kernels inside the PAC (Probably Approximately Correct) framework. From a more practical stand point, Boolean kernels have been successfully applied on many learning tasks, such as, face recognition [22,23], spam filtering [24], load forecasting [25], and on generic binary classification tasks [16,17].

## 3. Boolean Kernels for Categorical Data

Recently, a novel Boolean kernel framework for binary inputs, i.e., $x\in {\left\{0,1\right\}}^{n}$, has been proposed [8]. Differently from the kernels described in the previous section, this novel Boolean kernels family [8] defines feature spaces formed by specific types of Boolean formulas, e.g., only disjunctions of a certain degree or DNF with a specific structure. This framework offers a theoretically grounded approach for composing base Boolean kernels, namely conjunctive and disjunctive kernels, to construct kernels representing DNF formulas.

Specifically, the monotone conjunctive kernel (mC-kernel) and the monotone Disjunctive kernel (mD-kernel) of degree *d* between $x,z\in {\left\{0,1\right\}}^{n}$ are defined as follows: (1)κmCd= x⊺zd,(2)κmDd= nd−x⊺xd−z⊺zd+n−x⊺x−z⊺z+x⊺zd.

The mC-kernel counts the number of (monotone) conjunctions (formed using all different variables) involving *d* variables are satisfied in both x and z. Similarly, the mD-kernel computes the number of (monotone) disjunctions (formed using all different variables) involving *d* variables that are satisfied in both x and z. Boolean kernel functions heavily rely on the binomial coefficient because they consider clauses without repeated variables, e.g., $x_{1}\wedge x_{1}$ is not considered in the feature space of a mC-kernel of degree 2.

Starting from the mD- and mC-kernel (and their non-monotone counterpart) we can construct (m)DNF and (m)CNF kernels. The core observation is that DNFs are disjunctions of conjunctive clauses, and CNFs are conjunctions of disjunctive clauses. Thus, we can reuse the definitions of the mD- and mC-kernel to construct mDNF/mCNF kernels by replacing the dot-products (that represents the linear/literal kernel) with the kernel corresponding to the right Boolean operation. For example, if we want to construct an mDNF-kernel composed of three conjunctive clauses of degree 2 we need to substitute the dot-product in (Equation 2) with $\kappa_{\mathrm{mC}}^{2}$, obtaining
κmDNF3,2(x,z)= n23−x⊺x3−z⊺z3+n2−x⊺x−z⊺z+x⊺z3,
where the degree of the disjunction is 3, i.e., d=3, and the degree of the conjunctions is 2, i.e., c=2. The size of the input space for the disjunction becomes the size of the feature space of the mC-kernel of degree 2, i.e., n2. In its generic form, the mDNF-kernel(d, c) is defined as
κmDNFd,c(x,z)= ncd−x⊺xd−z⊺zd+nc−x⊺x−z⊺z+x⊺zd.

This kernel computes the number of mDNF formulas with exactly *d* conjunctive clauses of degree *c* that are satisfied in both x and z. This way of building (m)DNF and (m)CNF kernels imposes a structural homogeneity in the formulas, i.e., all elements in a clause have the same structure. Let us consider again the mDNF-kernel(3,2): each conjunctive clause has exactly two elements, and these elements are literals. As we will discuss in the next section, the BKs’ structural homogeneity is a limitation that can not be overcome using this framework.

Besides its theoretical value, this kernels family hugely improves the interpretability of Boolean kernels as shown in [9], and achieves state-of-the-art performance on both classification tasks [26] and on top-N item recommendation tasks [27].

In this work, we improve upon [8] by showing the limitations of this framework and providing a well-founded solution. The family of kernels presented here, called Propositional kernels, allows building Boolean kernels with feature spaces representing (almost) any logical proposition. In the remainder of this paper, we will use the term Boolean kernels to refer to the kernel family presented in [8].

### 3.1. Limitations of the Boolean Kernels

Before diving into the details of how to compute Propositional kernels, we show the limitations of the Boolean kernel framework [8], and how Propositional kernels overcome them.

Boolean kernels are designed to produce interpretable feature spaces composed of logical formulas over the input variables. However, the set of possible logical formulas that can be formed using Boolean kernels is limited. In particular, two aspects limit the logical expressiveness of the Boolean kernels [8]:(i)BKs do not consider clauses with the same variable repeated more than once (This refers to disjunctive and conjunctive BKs). Even though the features considered by BKs are, from a logical point of view, appropriate, they make the kernel definition cumbersome by introducing the binomial coefficient to generate all combinations without repetitions. For instance, the mC-kernel of degree 2 between $x,z\in {\left\{0,1\right\}}^{3}$ can be unfolded as
κmC2(x,z)= x⊺z2 =∑i∈[1,3]∑j∈[1,3]i≠j(xi∧xj)(zi∧zj)=(x1∧x2)(z1∧z2)+(x2∧x3)(z1∧z3)+(x2∧x3)(z2∧z3).Thus, the binomial coefficient is introduced to take into account only clauses with no repeated variables in it, e.g., avoiding clauses like $x_{2}\wedge x_{2}$.(ii)BKs are structurally “homogeneous”: each Boolean concept, described by a feature in the feature space, is homogeneous in their clauses. For example, an mDNF-kernel(3,2) creates mDNF formulas that are disjunctions of three conjunctive clauses of two variables. So, every single conjunctive clause is structurally homogeneous to the others (each of them has exactly 2 variables). It is not possible to form an mDNF of the form $\left(x_{1}\wedge x_{2}\wedge x_{3}\right)\vee \left(x_{1}\wedge x_{4}\right)$ where different conjunctive clauses have different degree.

For these reasons, in this paper, we propose a framework to produce kernels with a feature space that can potentially express any logical proposition over the input variables. To accomplish our goal, we need to overcome the limitations of the Boolean kernels, and we have also to provide a way to construct any possible logical formulas.

Overcoming the first limitation of the Boolean kernels, i.e., no repeated variables in the formulas, is very simple: it is sufficient to include any possible combination, even those that are logically a contradiction or a tautology, e.g., $x_{1}\wedge \neg x_{1}$ or $x_{1}\wedge x_{1}$. Regarding the homogeneity, some considerations need to be taken. Let us assume we want to create a kernel function such that its feature space is composed of monotone DNFs of the form f(a,b,c)=(a∧b)∨c, using the Boolean kernels. The embedding map of an mDNF-kernel [8] is defined as the composition of the embedding maps of the mD-kernel and the mC-kernel as:$${\tilde{\phi }}_{\mathrm{mDNF}}:x\mapsto {\tilde{\phi }}_{\mathrm{mD}}\left({\tilde{\phi }}_{\mathrm{mC}}\left(x\right)\right),\;x\in {\left\{0,1\right\}}^{n},$$
where we omitted the degrees and put a ~ over the functions to emphasize that we do not want to be linked to specific degrees. By this definition, there is no way to get a feature space with only formulas like *f* because we would need conjunctive clauses with different degrees, which is not feasible. Now, let say we redefine ${\tilde{\phi }}_{\mathrm{mC}}$, in such a way that it contains both conjunctions of degree 1 and degree 2, for instance, by summing an mC-kernel of degree 1 and an mC-kernel of degree 2 (the sum of two kernels induces a feature vector that is the concatenation of the kernels’ feature vectors). The resulting mapping ${\tilde{\phi }}_{\mathrm{mDNF}}$ would not create an embedding space with only *f*-like formulas anyway, because it would also contain formulas like a∨b. Unfortunately, we cannot overcome this last issue using Boolean kernels in the way they are defined.

The main problem originates from the basic idea behind Boolean kernels, that is creating logical formulas “reusing” the same set of inputs in each clause. Let us consider the simple case of a disjunctive kernel of degree 2. Given an input binary vector x, both literals in the disjunction are taken from x, thus they are by definition structurally identical (i.e., they are both literals). Now, let us consider an mDNF kernel with *d* disjunctions and conjunctive clauses of *c* literals. We have seen that an mDNF kernel is defined as the mD-kernel applied to the output of an mC-kernel. Hence, firstly, the conjunctive kernel embedding is computed, i.e., $\phi_{mC}^{c}\left(x\right)$, where all features have the same structure. Then, the disjunctive kernel embedding over $\phi_{mC}^{c}\left(x\right)$ is computed creating the final feature space. This case is the same as the previous example with the only difference that the input features are conjunctive clauses (with the same form) rather than literals. Thus, it is evident that by construction all the clauses are structurally homogeneous.

## 4. Propositional Kernels

Based on the observations made in the previous section, we now give the intuition behind the construction of Propositional kernels.

Let us take into consideration logical formulas of the form $f_{⊗}\left(a,b\right)=a⊗b$ where ⊗ is some binary Boolean operation over the variables a,b∈{0,1}. To construct formulas in such a way that *a* is taken from a set of variables and *b* from (possibly) another set, we need to consider two different input Boolean domains, which we call *A* and *B*, respectively. These domains are generally intended as sets of other Boolean formulas over the input variables. Now, given an input vector $x\in {\left\{0,1\right\}}^{n}$, we map x in both the domain *A* and *B*, i.e., $\phi_{A}\left(x\right)$ and $\phi_{B}\left(x\right)$, and then we perform the Boolean function $f_{⊗}$ by taking one variable from $\phi_{A}\left(x\right)$ and one from $\phi_{B}\left(x\right)$. Figure 1 graphically shows the just described procedure.

Formally, we can define a generic propositional embedding function for the logical operator ⊗ over the domains *A* and *B* as:(3)$$\phi_{A⊗B}\left(x\right):x\mapsto {\left(\phi_{A}{\left(x\right)}_{a}⊗\phi_{B}{\left(x\right)}_{b}\right)}_{a\in A,b\in B},$$
and consequently the corresponding kernel function $\kappa_{A⊗B}\left(x,z\right)$ is defined by
(4)$$\kappa_{A⊗B}\left(x,z\right)=\langle \phi_{A⊗B}\left(x\right),\phi_{A⊗B}\left(z\right)\rangle =\sum_{(a,b)\in A\times B}\left(\phi_{A}{\left(x\right)}_{a}⊗\phi_{B}{\left(x\right)}_{b}\right)\left(\phi_{A}{\left(z\right)}_{a}⊗\phi_{B}{\left(z\right)}_{b}\right),$$
with $x,z\in {\left\{0,1\right\}}^{n}$. The kernel $\kappa_{A⊗B}\left(x,z\right)$ counts how many logical formulas of the form a⊗b, with *a* taken from the feature space of $\phi_{A}$ and *b* taken from the feature space of $\phi_{B}$, are satisfied in both x and z. To check whether this formulation is ideally able to build kernels for a generic Boolean formula, let us reconsider the example of the previous section, i.e., f(a,b,c)=(a∧b)∨c. If we assume that the domain *A* contains all the formulas of the form (a∧b), while the domain *B* contains single literals (actually it corresponds to the input space), then by using Equation (Equation 4) we can define a kernel for *f* by simply posing ⊗≡∨. In the next section, we expand upon the idea showed in the example above proving that we can implement any propositional formula that not contains bound literals (we will discuss this limitation). However, we need to design a method to compute it without expliciting any of the involved spaces (except for the input space).

### 4.1. Construction of a Propositional Kernel

Since we want to be as general as possible, we need to define a constructive method for generating and computing a Propositional kernel, rather than a specific formulation for any possible formula. To do this, we use the fact that Boolean formulas can be defined as strings generated by a context-free grammar.

**Definition** **1**(Grammar for propositional logic [28]). *Formula in propositional logic are derived from the context-free grammar, $G_{P}$, whose terminals are:*
*a set of symbols P called atomic propositions;**a set B of Boolean operators.*
*The context-free grammar $G_{P}$ is defined by the following productions:*
$$\begin{array}{cc} F & ::=p,\;p\in P\\ F & ::=\neg \;F\\ F & ::=F⊗F,\;⊗\in B \end{array}$$

*A formula is a word that can be derived from the non-terminal F.*


Starting from the grammar $G_{P}$, we can define the Propositional kernel framework by providing a kernel for each production (that we call “base” Propositional kernels), and then, with simple combinations, we can build any Propositional kernel by following the rules of the grammar.

The first production, i.e., F::=p, is trivial since is the literal kernel (or monotone literal kernel in the Boolean kernels’ jargon), i.e., the linear kernel. Similarly, the second production, i.e., F::=¬F, which represents the negation, corresponds to the NOT kernel (or negation kernel for Boolean kernels), $\kappa_{\neg }$, which is simply the linear kernel between the “inverse” of the input vectors, i.e., $\kappa_{\neg }\left(x,z\right)={\left(1_{n}-x\right)}^{⊺}\left(1_{n}-z\right)$, where $1_{n}$ is an *n*-dimensional vector with all entries equal to 1.

The third and last production, i.e., F::=F⊗F, represents a generic binary operation between logical formulas and/or literals. To be general with respect to the operation F⊗F, we need to distinguish the two operands and we cannot make any assumption about what the operator represents. For these reasons, we will refer to the first and the second operand with *A* and *B*, respectively. Regarding the operation ⊗, we consider a generic truth table as in Table 1, where $y_{⊗}\left(a,b\right)$ is the truth value given all possible combinations of a,b∈{0,1}.

The kernel we want to define for the operation ⊗ has exactly the form of the kernel $\kappa_{A⊗B}\left(x,z\right)$ previously described (Equation (Equation 4)): each operand is taken from (potentially) different spaces, i.e., different formulas, and the kernel counts how many of these logical operations are satisfied in both the input vectors.

Since we have to count the common true formulas, we need to take into account the configurations of *a* and *b* that generate a true value for $y_{⊗}\left(a,b\right)$, and given those true configurations we have to consider all the true joint configurations between the inputs. In other words, a formula can be true for x for a certain configuration while it can be also true for z for another configuration. For instance, let the formula be a disjunction of degree 2, i.e., a∨b. Then, given a feature of the embedding space, this can be true for x because its *a*-part is true, and vice versa for z. To clarify this last concept, please consider Table 2.

It is evident that the value of the kernel is the sum over all the possible joint configurations of the common true target values between x and z. To compute this, for each row of the Table 2 we calculate the true formulas in x and z for the configuration corresponding to the row. For example, in the first row of the table we have to count all the common true formulas such that the features in *A* and in *B* are false in both x and z, and this can be computed by:$${\left(1_{\left|A\right|}-\phi_{A}\left(x\right)\right)}^{⊺}\left(1_{\left|A\right|}-\phi_{A}\left(z\right)\right){\left(1_{\left|B\right|}-\phi_{B}\left(x\right)\right)}^{⊺}\left(1_{\left|B\right|}-\phi_{B}\left(z\right)\right),$$
which is actually the product of the NOT kernels in the domain *A* and *B*, that is:$$\kappa_{\neg }\left(\phi_{A}\left(x\right),\phi_{A}\left(z\right)\right)\kappa_{\neg }\left(\phi_{B}\left(x\right),\phi_{B}\left(z\right)\right)=\kappa_{\neg A}\left(x,z\right)\kappa_{\neg B}\left(x,z\right).$$

Such computation can be generalized over all the possible 16 configurations, i.e., the rows of the joint truth table, by the following formula
(5)$$\kappa_{A⊗B}\left(x,z\right)=\sum_{(a_{x},b_{x})\in B}\sum_{(a_{z},b_{z})\in B}y_{⊗}\left(a_{x},b_{x}\right)y_{⊗}\left(a_{z},b_{z}\right)\Psi_{A}{\left(x,a_{x}\right)}^{⊺}\Psi_{A}\left(z,a_{z}\right)\Psi_{B}{\left(x,a_{x}\right)}^{⊺}\Psi_{B}\left(z,b_{z}\right)$$
where B≡{(a,b)∣a∈{0,1},b∈{0,1}} and $\Psi_{A}:{\left\{0,1\right\}}^{n}\times \left\{0,1\right\}\to {\left\{0,1\right\}}^{\left|A\right|}$ is defined as
$$\Psi_{A}\left(x,a_{x}\right)=\left(1-a_{x}\right)1_{\left|A\right|}+\left(2a_{x}-1\right)\phi_{A}\left(x\right)=\;\left\{\begin{array}{cc} \phi_{A}\left(x\right) & \mathrm{if}\;a_{x}=1\\ 1_{\left|A\right|}-\phi_{A}\left(x\right) & \mathrm{if}\;a_{x}=0 \end{array}\right.,$$
and the definition is analogous for $\Psi_{B}$.

In its worst case, that is when the joint truth table has 1 in every configuration, the formula has 16 non-zero terms. However, we have to underline that only a small set of operations need the computation of the corresponding kernel via Equation (Equation 5) since we can use logic equivalences and apply them with the Propositional kernels. The only exceptions where the logic equivalences do not hold for the Propositional kernels is when there are constraints in the variables. For example, in logic we can express the xor operation by means of *and*, *or* and *not*, i.e., a⊕b↔(a∧¬b)∨(¬a∧b), but this cannot be done with kernels since we have no way to fix the relations between the first conjunctive clause and the second conjunctive clause. It is worth noticing that this is expected behavior since the Propositional kernels have a feature space with all possible arrangements of the variables in the target formula. Indeed, we can define a Propositional kernel that represents the formula (a∧¬b)∨(¬c∧d) which contains the *xor*-equivalent formula (when a=c and b=d) along with all other formulas of that form. Still, the Propositional *XOR* kernel can be defined but not using the definition above (see next section). In practical terms, this limitation should not be critical since we are unaware of which type of formulas is needed and it makes sense to try with generic formulas than constrained ones. In all the other cases, logic equivalences hold, e.g., the De Morgan’s laws and the double negation rule, and this allows us to compute, for example, the implication kernel in terms of the disjunctive Propositional kernel and the NOT kernel.

In the following we provide a couple of examples of Equation (Equation 5) for computing the Propositional kernels.

**Example** **1**(Conjunction). *The truth table of the conjunction has only one true output, that is $y_{\wedge }\left(1,1\right)$. Hence, there exists a unique term in the summation of $\kappa_{A\wedge B}$ s.t. $y_{\wedge }\left(a_{x},b_{x}\right)\cdot y_{\wedge }\left(a_{z},b_{z}\right)=1$, that is when $a_{x}=b_{x}=a_{z}=b_{z}=1$. This leads to the following formulation*
$$\begin{array}{cc} \kappa_{A\wedge B}\left(x,z\right) & =\Psi_{A}{\left(x,1\right)}^{⊺}\Psi_{A}\left(z,1\right)\;\cdot \;\Psi_{B}{\left(x,1\right)}^{⊺}\Psi_{B}\left(z,1\right)\\ \; & =\phi_{A}{\left(x\right)}^{⊺}\phi_{A}\left(z\right)\;\cdot \;\phi_{B}{\left(x\right)}^{⊺}\phi_{B}\left(z\right)\\ \; & =\kappa_{A}\left(x,z\right)\kappa_{B}\left(x,z\right), \end{array}$$*which is actually the number of possible conjunctions a∧b that are satisfied in both x and z s.t. a∈A,b∈B. This can be defined as the product between the number of common true formulas in A and the number of common true formulas in B, that is the product of the kernels $\kappa_{A}$ and $\kappa_{B}$.*

**Example** **2**(Exclusive disjunction). *The truth table of the exclusive disjunction has two true outputs, that is when a and b have different truth values. So in this case, the joint truth table have four non-zero terms:*
$$\begin{array}{cc} \kappa_{A⊕B}\left(x,z\right)= & \Psi_{A}{\left(x,1\right)}^{⊺}\Psi_{A}\left(z,1\right)\;\cdot \;\Psi_{B}{\left(x,0\right)}^{⊺}\Psi_{B}\left(z,0\right)+\\ \; & \Psi_{A}{\left(x,1\right)}^{⊺}\Psi_{A}\left(z,0\right)\;\cdot \;\Psi_{B}{\left(x,0\right)}^{⊺}\Psi_{B}\left(z,1\right)+\\ \; & \Psi_{A}{\left(x,0\right)}^{⊺}\Psi_{A}\left(z,1\right)\;\cdot \;\Psi_{B}{\left(x,1\right)}^{⊺}\Psi_{B}\left(z,0\right)+\\ \; & \Psi_{A}{\left(x,0\right)}^{⊺}\Psi_{A}\left(z,0\right)\;\cdot \;\Psi_{B}{\left(x,1\right)}^{⊺}\Psi_{B}\left(z,1\right), \end{array}$$*that through simple math operations is equal to*
$$\begin{array}{cc} \kappa_{A⊕B}\left(x,z\right)=\; & \kappa_{A}\left(x,z\right)\kappa_{\neg B}\left(x,z\right)+\\ \; & \left(\kappa_{A}\left(x,x\right)-\kappa_{A}\left(x,z\right)\right)\left(\kappa_{B}\left(z,z\right)-\kappa_{B}\left(x,z\right)\right)+\\ \; & \left(\kappa_{A}\left(z,z\right)-\kappa_{A}\left(x,z\right)\right)\left(\kappa_{B}\left(x,x\right)-\kappa_{B}\left(x,z\right)\right)+\\ \; & \kappa_{\neg A}\left(x,z\right)\kappa_{B}\left(x,z\right), \end{array}$$*where $\kappa_{\neg A}$ and $\kappa_{\neg B}$ are the NOT kernels applied on the domains A and B, respectively.*

### 4.2. Propositional Kernels’ Definition

In this section, we will refer to $\kappa_{A}$ and $\kappa_{B}$ as two generic kernel functions of the type ${\left\{0,1\right\}}^{n}\times {\left\{0,1\right\}}^{n}\to N$, such that $\kappa_{A}\left(x,z\right)=\langle \phi_{A}\left(x\right),\phi_{A}\left(z\right)\rangle$ and $\kappa_{B}\left(x,z\right)=\langle \phi_{B}\left(x\right),\phi_{B}\left(z\right)\rangle$, where $\phi_{A}:{\left\{0,1\right\}}^{n}\to {\left\{0,1\right\}}^{n_{A}}$ and $\phi_{B}:{\left\{0,1\right\}}^{n}\to {\left\{0,1\right\}}^{n_{B}}$.

Using the procedure shown in the previous section and simple mathematical/logical simplification we can provide a slimmer kernel definition for all 16 possible truth tables. Table 3 summarizes the definition of the Propositional kernels for all the unary and binary logical operations. Note that in the table the kernels corresponding to the following propositions are missing: *⊤* (logical true), *⊥* (logical false), b→a, and b¬→a. Both *⊤* and *⊥* are not included because they are defined as the constant matrix 1 and I, respectively. While for b→a and b¬→a it is sufficient to swap the role of *A* and *B* in the provided definitions for a→b and a¬→b. Finally, both the literal and the not operator are the same for *B* but in the table we report only the definition for *A*.

### 4.3. Relation with Boolean Kernels

The main difference between Boolean kernels [8] and Propositional kernels have already been discussed, i.e., Boolean kernels have a homogeneous structure and Propositional kernels do not. This difference implies that there exists a discrepancy between the same type of kernels of the two families. For instance, consider the simple conjunction a∧b, and compare the Boolean monotone conjunctive kernel ($\kappa_{mC}$) with the propositional AND kernel ($\kappa_{\wedge }$):κ∧(x,z)=(x⊺z)2,κmC(x,z)= x⊺z2 =12(x⊺z)(x⊺z−1).

The core difference lies in the combinations of features considered by the two kernels. On one hand, $\kappa_{mC}$ only considers all the combination without repetition of the input variables, thus excluding conjunctions like $x_{1}\wedge x_{1}$ and counting only once the conjunction $x_{1}\wedge x_{2}$. On the other hand, $\kappa_{\wedge }$ takes into consideration all the possible pairs of variables, including also $x_{1}\wedge x_{1}$ and both $x_{1}\wedge x_{2}$ and $x_{2}\wedge x_{1}$. Albeit Propositional kernels are computed on “spurious” combinations, this is indeed crucial to allow the overall framework to work and overcomes the limitations of Boolean kernels.

## 5. Propositional Kernels’ Composition

All the definitions provided in Table 3 can be used as building blocks to construct kernels for any propositional formula. Specifically, we use the grammar $G_{P}$ to construct the parsing tree of the target propositional formula and then we apply the kernels defined in Table 3 in a bottom-up fashion starting from the leaves of the parsing tree up to the root.

Leaves, that correspond to the only terminal production in the grammar $G_{P}$, are always the literal kernel. Then, moving up to the tree any of the other "base" Propositional kernels can be encountered, and the corresponding kernel is applied where the generic domain symbols *A* and/or *B* are substituted with the domain (i.e., logical proposition) resulting from the tree rooted in the current node. Let us explain this procedure through an example.

**Example** **3**(Constructing the Propositional kernel for the proposition f=(a∧¬b)∨¬c).
*The construction of the kernel follows the structure of the parsing tree, depicted in Figure 2, from the leaves to the root. The leaves, as previously said, are literal kernels between the input vectors, thus it is the linear kernel, i.e., $\kappa_{A}\left(x,z\right)=\kappa_{B}\left(x,z\right)=\kappa_{C}\left(x,z\right)=x^{⊺}z$. Then, the NOT kernel is applied on both b and c, $\kappa_{\neg B}\left(x,z\right)=\kappa_{\neg C}\left(x,z\right)=n+x^{⊺}z-x^{⊺}x-z^{⊺}z$. Afterwards, the left branch applies the AND kernel between the input a and the latent representation of the NOT kernel on b, i.e., $\kappa_{A\wedge \neg B}\left(x,z\right)=\left(x^{⊺}z\right)\left(n+x^{⊺}z-x^{⊺}x-z^{⊺}z\right)$. Finally, the root applies the OR kernel between the representations of the two branches. The OR kernel is defined in terms of AND kernels applied to the negation of the two expressions involved in the disjunction, in this case (a∧¬b) and ¬c.*

*Let us start by computing $\kappa_{\neg \neg C}$:*
$$\begin{array}{cc} \kappa_{\neg \neg C}\left(x,z\right) & =\kappa_{\neg (\neg C)}\left(x,z\right)\\ \; & =n+\left(n+x^{⊺}z-x^{⊺}x-z^{⊺}z\right)-\left(n+x^{⊺}x-x^{⊺}x-x^{⊺}x\right)-\left(n+z^{⊺}z-z^{⊺}z-z^{⊺}z\right)\\ \; & =x^{⊺}z-x^{⊺}x-z^{⊺}z+x^{⊺}x+z^{⊺}z\\ \; & =x^{⊺}z=\kappa_{C}\left(x,z\right). \end{array}$$

*Unsurprisingly, computing $\kappa_{\neg \neg C}$, is equivalent to compute $\kappa_{C}$, that shows that the logical double negation rule applies to Propositional kernels. Now, for computing $\kappa_{\neg (A\wedge \neg B)}$, we show that the De Morgan’s law also holds:*
$$\begin{array}{cc} \kappa_{\neg (A\wedge \neg B)}\left(x,z\right)=n^{2} & +\left(x^{⊺}z\right)\left(n+x^{⊺}z-x^{⊺}x-z^{⊺}z\right)\\ \; & -\left(x^{⊺}x\right)\left(n-x^{⊺}x-x^{⊺}x-x^{⊺}x\right)\\ \; & -\left(x^{⊺}z\right)\left(n-z^{⊺}z-z^{⊺}z-z^{⊺}z\right)=\kappa_{\neg A\vee B}\left(x,z\right)=\kappa_{A\to B}\left(x,z\right) \end{array}$$

*Finally, we put everything together, obtaining:*
$$\kappa_{f}\left(x,z\right)=n^{3}+\kappa_{A\to B}\left(x,z\right)\kappa_{C}\left(x,z\right)-\kappa_{A\to B}\left(x,x\right)\kappa_{C}\left(x,x\right)-\kappa_{A\to B}\left(z,z\right)\kappa_{C}\left(z,z\right).$$


Algorithm 1 shows the pseudo-code of the just-described procedure. In the algorithm, $K_{\neg }$ is a function with signature $\left(K_{A},|A|\right)$ with $K_{A}$ a Propositional kernel matrix and |A| the dimension of its feature space. Similarly, in the binary case, the functions’ signatures are $\left(K_{A},|A|,K_{B},|B|\right)$ with $K_{A},K_{B}$ Propositional kernel matrices and |A|,|B| the dimension of their feature spaces. All the kernel functions return a tuple made up of a kernel matrix and its feature space dimension.

The source code is available at https://github.com/makgyver/propositional_kernels (accessed on 6 August 2021).

**Algorithm 1:** compute_kernel: compute a Propositional kernel (for a data set **X**) given a propositional parsing tree.

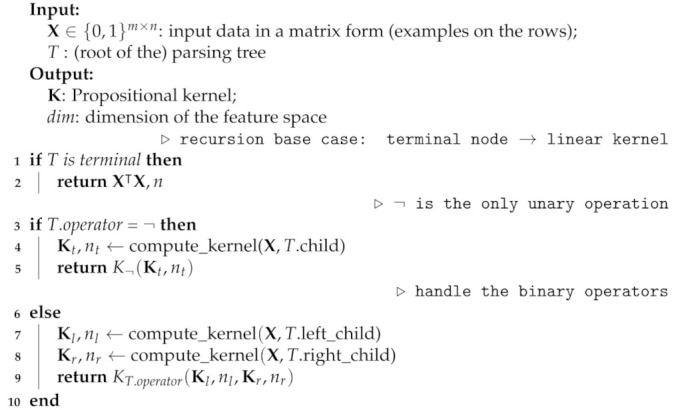



## 6. Propositional Kernels’ Application

In this section, we evaluate Propositional kernels on binary classification tasks. Specifically, we conducted two sets of experiments: one on artificial data sets and one on benchmark data sets. In the first set of experiments, we assessed the benefit of using Propositional kernels on artificially data sets created with the following procedure:(1)Generate a binary matrix (with examples on the rows) $X\in {\left\{0,1\right\}}^{m\times n}$ where each row represents a distinct assignment of the *n* binary variables (i.e., $m=2^{n}$);(2)Generate a random logical proposition *f* over the *n* variables using Algorithm 2;(3)Create the vector of labels $y\in {\left\{0,1\right\}}^{m}$ such that $y_{i}=1$ iff $f(x_{i})=⊤$, 0 otherwise.

This “controlled” test aims at showing that the Propositional kernel corresponding to the rule that generates the labels guarantees good classification performance.

**Algorithm 2:** generate_rule: Algorithm for generating the parsing tree of a random propositional rule.

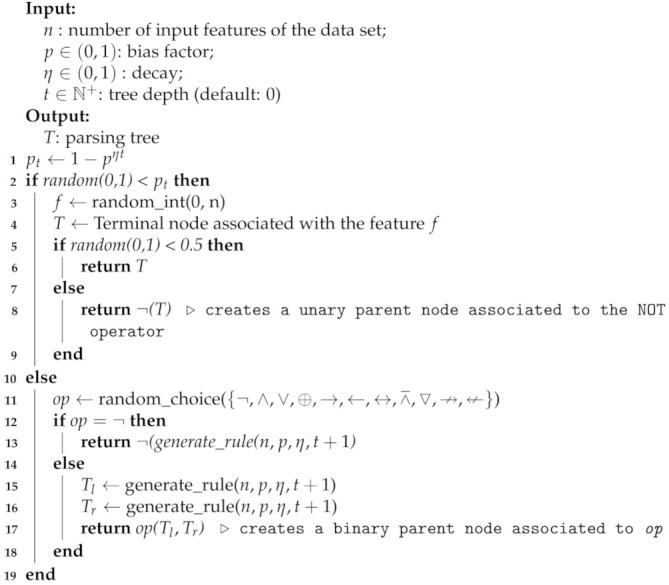



In these experiments we created the artificial data sets using Algorithm 2 with η=0.7 and p=0.4. We chose these values to obtain relatively short formulas that explain the labels. Figure 3 shows the distribution of the length (i.e., number of leaves in the parsing tree) of the formulas over 1000 generations.

As evident from the figure, the used parametrization gives a bias towards short rules (i.e., 2 or 3 literals), and the probability of having longer rules quickly decays. In terms of explainability, shorter rules are easier to understand than longer and more complex ones. For this reason, we perform this set of experiments using this rules length distribution.

Given the data set (X,y), we compared the AUC of the Propositional kernel implementing the rule that generated y (called *Target kernel*) against 10 Propositional kernels that implement a randomly generated rule using the same approach (with the same parameters) described in Algorithm 2. The used kernel machine is a hard-margin SVM and n=10 which generates a total of 1024 (=$2^{10}$) examples of which 30% has been held out as the test set. We performed the comparison varying the size of the training set and the overall procedure has been repeated 100 times. The average results over all the repetitions are reported in Figure 4.

The plots show that the target kernel can achieve AUC > 0.99 with only 100 examples (i.e., ∼10% of the all data set), and it achieves almost perfect classification accuracy with 200 examples. The performance of the other Propositional kernels remains on average a 3% behind, and never achieves perfect classification despite using all the training examples. These results show that using a Propositional kernel implementing the rule governing the labeling ensure high classification accuracy.

However, in real classification problems, we do not know beforehand if and which rule over the input variables produces the labels. Thus, we propose a simple yet efficient heuristic to select an appropriate Propositional kernel while avoiding performing an extensive validation. Given its combinatorial nature, we cannot systematically generate all possible Propositional kernels. Moreover, assuming that we can select a subset of possible good Propositional kernels, performing validation over all such kernels could be very demanding especially on large-scale data sets.

Thus, our proposal is based on two desiderata:1.we are interested in generating kernels that can be helpful to interpret the solution of a kernel machine;2.we want to avoid to perform an extensive and time-consuming validation.

To cope with our first goal, we generate kernels using the same procedure used in the previous set of experiments, i.e., kernel-based on rules generated by Algorithm 2 with the same parameters. In this way, we may enforce a bias toward formulas of shorter length for ensuring decent explainability. Nonetheless, by acting on *p* and η we can also compute very complex formulas if we do not care much about explainability. On one hand, by acting on the value of *p*, is possible to shift the peak of the distribution towards longer (p→1) or shorter (p→0) rules (Figure 5). On the other hand, by changing η is possible to change the range of the degree. High values of η decrease the range, while smaller values (η→0) would allow having rules with hundreds of literals (Figure 5).

Moreover, Algorithm 2 can be substituted with any other procedure that randomly generates rules that are then used to compute the kernels. The kernel generation procedure should be designed to leverage any a-priori knowledge about the data set at hand. However, in these experiments, we assume no prior knowledge about the data sets, and the provided bias aims at providing (potentially) effective but easy to explain rules.

To avoid the validation procedure, we suggest selecting, from the pool of generated kernels, the one that minimizes the radius-margin ratio [29,30,31,32], that is the ratio between the radius of the minimum enclosing ball that contains all the training examples and the margin observed on training data. A similar strategy has been previously employed in [30,32]. The intuition is that by minimizing the radius-margin ratio we aim at selecting a representation with a large margin and small radius that we can expect to achieve a tighter generalization error bound which can lead to better performance. To validate this procedure, we performed a set of experiments on 10 categorical benchmark data sets which details are reported in Table 4.

In these experiments, we used a soft-margin SVM. We validated the hyper-parameter *C* in the set $\left\{10^{-5},\cdots ,10^{4}\right\}$ using nested 5-fold cross-validation, and all kernels have been normalized, that is, given a kernel matrix K we computed its normalized version $\tilde{K}$ as$$\tilde{K}=\frac{K}{\sqrt{dd^{⊺}}},$$where d is a *n*-dimensional vector containing the diagonal of K.

Given a data set, we randomly generated 30 Propositional kernels and for each of them, we trained a soft-margin SVM. In Figure 6, we show, for each data set, the achieved AUC w.r.t. the radius-margin ratio of each of the 30 kernels. As a reference, we also indicate with a red dotted line the performance of the linear kernel as well as the performance (blue dotted line) of the best performing Boolean Kernel according to [8]. We tested all monotone and non-monotone Boolean kernels (i.e., disjunctive, conjunctive, CNF and DNF) up to degree 5 for both the conjunctive and the disjunctive clauses. The best Boolean kernel is selected according to the AUC on the test set.

From the plots, and from the correlation analysis reported in Table 5 it is evident that the minimal radius-margin ratio represents a good, although not perfect, heuristic. There are three cases in which the correlation between low radius-margin ratio and high AUC is not statistically significant, namely for house-votes, spect and splice. However, in these data sets, it seems that almost all of the randomly generated Propositional kernels have a similar performance which makes the heuristic not useful but still not harmful. There are also two data sets (monks and primary-tumor) in which the negative correlation is weakly significant, yet the kernel with the lowest radius-margin ratio is competitive: best performance in monks and ∼0.5% inferior to the best AUC in primary-tumor.

In almost all data sets the linear kernel performs badly with respect to the Propositional kernels. Nonetheless, on both spect and house-votes the linear kernel performs decently. In particular, on spect the generated Propositional kernels seem to be sub-optimal and the best performance is achieved by Boolean kernels. We argue that in this specific data set kernels with low complexity (e.g., low spectral-ratio [31]) tend to perform better. Indeed, the tested Boolean kernels implement simple combinations of disjunctions and conjunctions, while Propositional kernels are generally more complex given the wider range of operations that they can implement. This is further supported by the good performance of the linear kernel.

Despite the just mentioned exception, Propositional kernels achieve the highest AUCs in all other data sets exceeding 98% of AUC in all but primary-tumor. This particular classification task seems to be more difficult than the others and this is further underlined by the poor performance of both the linear and the best Boolean kernel.

It is also noteworthy that in all data sets, except primary-tumor, promoters and soybean, the radius-margin ratio achieved by the linear kernel was out of scale w.r.t. to the ones in the plots. This is due to the fact that such data sets are not linearly separable.

## 7. Conclusions and Future Work

We proposed a novel family of kernels, dubbed Propositional kernels, for categorical data. The Propositional kernel framework improves upon Boolean kernels, overcoming their expressiveness’ limitations. We show how Propositional kernels can be built and composed providing the python source code. An extensive empirical evaluation on artificial and benchmark categorical data sets demonstrates the effectiveness of this new family of kernels.

The natural follow-up of this work is to use Propositional kernels to explain the decision of an SVM. Given their expressivity richness, Propositional kernels may allow generating shorter and in general more accurate rules than Boolean kernels. Another direction that might be explored is to use Multiple Kernel Learning (MKL) combining different Propositional kernels. This would allow combining a set of Propositional kernels leaving to the MKL approach the task of selecting (i.e., weighting) the most relevant kernels for the task at hand. 

## Figures and Tables

**Figure 1 entropy-23-01020-f001:**
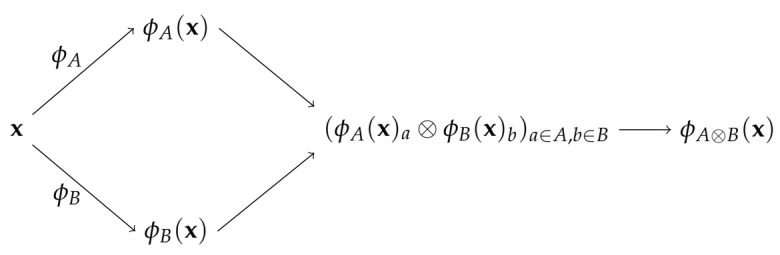
A graphical depiction of the idea behind Propositional kernels for breaking the homogeneity of Boolean kernels: the input vector is mapped onto two, potentially different, spaces, and then the final feature space is composed of all possible pairs of features one taken from the space of $\phi_{A}$ and one taken from the space of $\phi_{B}$. Finally the logical interpretation of such final features depends on ⊗.

**Figure 2 entropy-23-01020-f002:**
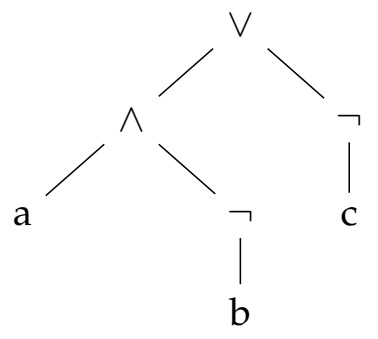
Parsing tree for (a∧¬b)∨¬c.

**Figure 3 entropy-23-01020-f003:**
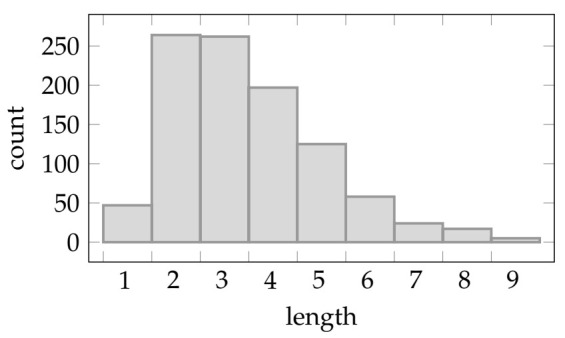
Distribution of the rules length (i.e., number of literals) on 1000 random Propositional kernel generations with p=0.4 and η=0.7 (i.e., the values used in the experiments).

**Figure 4 entropy-23-01020-f004:**
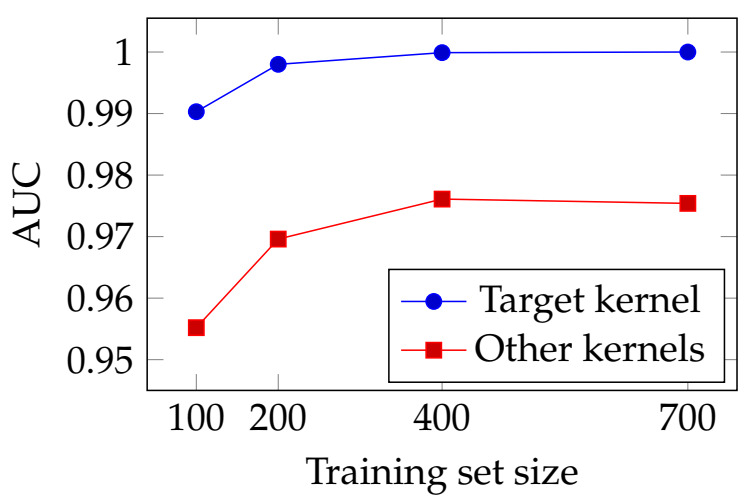
AUC results varying the number of training examples.

**Figure 5 entropy-23-01020-f005:**
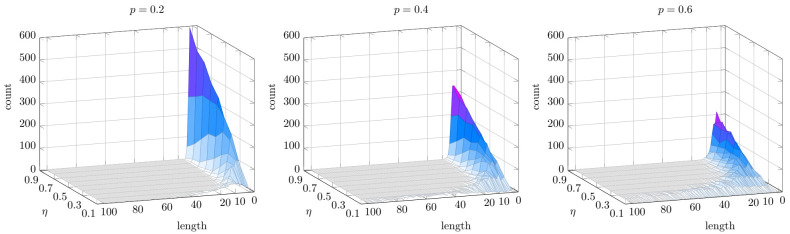
Distribution of the rules length on 1000 randomly generated Propositional kernels varying p∈{0.2,0.4,0.6} and η∈{0.1,⋯,0.9}. Higher values of *p* are not included because the generative procedure becomes too slow.

**Figure 6 entropy-23-01020-f006:**
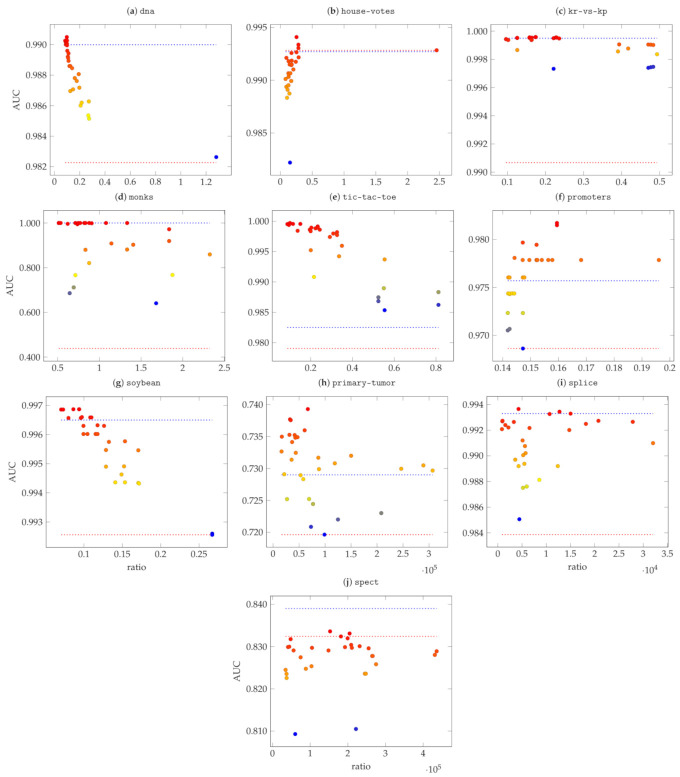
AUC score and reached ratio for each considered data set. In general, high AUC values correspond to small ratio. The red dotted line indicates the performance of the linear kernel, while the blue dotted line is the highest AUC achieved by a Boolean kernel on the test set. The color of the dots indicate the relative performance w.r.t. the others, namely red is the highest, blue the lowest.

**Table 1 entropy-23-01020-t001:** Generic truth table for the ⊗ operation.

*a*	*b*	$y_{⊗}\left(a,b\right)$
0	0	0⊗0
0	1	0⊗1
1	0	1⊗0
1	1	1⊗1

**Table 2 entropy-23-01020-t002:** Joint truth table between x and z for the operation ⊗.

$a_{x}$	$b_{x}$	$a_{z}$	$b_{z}$	$y_{⊗}\left(a_{x},b_{x}\right)y_{⊗}\left(a_{z},b_{z}\right)$
0	0	0	0	(0⊗0)(0⊗0)
0	1	0	0	(0⊗1)(0⊗0)
1	0	0	0	(1⊗0)(0⊗0)
1	1	0	0	(1⊗1)(0⊗0)
0	0	0	1	(0⊗0)(0⊗1)
⋮	⋮	⋮	⋮	⋮
1	1	1	1	(1⊗1)(1⊗1)

**Table 3 entropy-23-01020-t003:** “Base” Propositional kernels definition.

κ	Logic Op.	Name	κ(x,z)	|κ|
$\kappa_{A}$	*a*	literal	$\kappa_{A}\left(x,z\right)$	$n_{A}$
$\kappa_{\neg A}$	¬a	not	$n_{A}+\kappa_{A}\left(x,z\right)-\kappa_{A}\left(x,x\right)-\kappa_{A}\left(z,z\right)$	$n_{A}$
$\kappa_{A\wedge B}$	a∧b	and	$\kappa_{A}\left(x,z\right)\kappa_{B}\left(x,z\right)$	$n_{A}n_{B}$
$\kappa_{A\vee B}$	a∨b	or	$n_{A}n_{B}-\kappa_{\left(\neg A)\wedge (\neg B\right)}\left(x,x\right)-\kappa_{\left(\neg A)\wedge (\neg B\right)}\left(z,z\right)+\kappa_{\left(\neg A)\wedge (\neg B\right)}\left(x,z\right)$	$n_{A}n_{B}$
			$2\kappa_{A\wedge B}\left(x,z\right)+\kappa_{A\wedge (\neg B)}\left(x,z\right)+\kappa_{(\neg A)\wedge B}\left(x,z\right)+$	
$\kappa_{A⊕B}$	a⊕b	xor	$-\kappa_{A}\left(x,z\right)\left(\kappa_{B}\left(x,x\right)+\kappa_{B}\left(z,z\right)\right)-\kappa_{B}\left(x,z\right)\left(\kappa_{A}\left(x,x\right)+\kappa_{A}\left(z,z\right)\right)+$	$n_{A}n_{B}$
			$+\kappa_{A}\left(x,x\right)\kappa_{B}\left(z,z\right)+\kappa_{B}\left(x,x\right)\kappa_{A}\left(z,z\right)$	
$\kappa_{A\to B}$	a→b	implication	$\kappa_{(\neg A)\vee B}\left(x,z\right)$	$n_{A}n_{B}$
$\kappa_{A↔B}$	a↔b	equivalence	$\kappa_{\neg (A⊕B)}\left(x,z\right)$	$n_{A}n_{B}$
$\kappa_{A\neg \to B}$	a¬→b	not implicat.	$\kappa_{\neg ((\neg A)\vee B)}\left(x,z\right)=\kappa_{A\wedge (\neg B)}\left(x,z\right)$	$n_{A}n_{B}$
$\kappa_{A⊼B}$	a⊼B	nand	$\kappa_{\neg (A\wedge B)}$	$n_{A}n_{B}$
$\kappa_{A▿B}$	a▿B	nor	$\kappa_{\neg (A\vee B)}$	$n_{A}n_{B}$

**Table 4 entropy-23-01020-t004:** Datasets information: name, number of instances, number of features, classes distribution and number of active variables (ones) in every example (i.e., ${∥x∥}_{1}$).

Dataset	#Instances	#Features	Distribution (%)	${∥x∥}_{1}$
dna	2000	180	47/53	47
house-votes	232	32	47/53	16
kr-vs-kp	3196	73	52/48	36
monks	432	17	33/67	6
primary-tumor	339	34	42/58	15
promoters	106	228	50/50	57
soybean	266	97	55/45	35
spect	267	45	79/21	22
splice	3175	240	48/52	60
tic-tac-toe	958	27	65/35	9

**Table 5 entropy-23-01020-t005:** Pearson correlation coefficient (with associated *p*-value) between the radius-margin ratio and the AUC of the generated Propositional kernels. Negative correlation means that lower values of radius-margin ratio are correlated to higher AUC values. The column “Significance” indicates with a 4 star-based notation the significance of the results, where * ≡p∈[0.1,0.01), ** ≡p∈[0.01,0.001), *** ≡p∈[0.001,0.0001), and **** ≡p≤0.0001.

Dataset	Pearson	*p*-Value	Significance
splice	0.195	3.19 × 10${}^{-1}$	-
soybean	−0.931	8.74 × 10${}^{-14}$	****
tic-tac-toe	−0.852	8.90× 10${}^{-9}$	****
promoters	0.4634	9.92× 10${}^{-3}$	**
house-votes-84	0.2483	1.94× 10${}^{-1}$	-
dna	−0.7401	6.75× 10${}^{-6}$	****
spect	0.1265	5.05× 10${}^{-1}$	-
kr-vs-kp	−0.6239	3.89× 10${}^{-4}$	***
primary-tumor	−0.2896	1.00× 10${}^{-1}$	*
monks-1	−0.3008	1.00× 10${}^{-1}$	*

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
