# Peer review of "Propositional Kernels"

_entropy, 2021, doi:10.3390/e23081020_

Round 1

Reviewer 1 Report

This paper extends the previous works of this group on Boolean kernels to what they call propositional kernels. While the former were limited to essential Boolean operations such as AND or OR, the latter are now able to compose arbitrary Boolean functions from a kernel truth table of 16 different base functions.

The content constitutes a natural next step in this line of work, namely the combination of logic and continuous learning, in this case SVMs.

My main concern is in the experiments (Figure 3):

  1. I would have liked to see the performance of Boolean kernels. Different logical functions are compared to linear kernels (and often outperform them), but how well would the Boolean kernels do? I really think that this is necessary for estimating the value of the propositional extension.
  2. A nice feature of the algorithm is the efficient model selection based on the radius-margin ratio. The authors conclude from Fig. 3 that a low value of the ratio is good. Frankly, I don't really see this. There are some datasets where this is obviously the case (e.g., soybean or dna), but there are others where it seems to be exactly the opposite (such as promoters or splice), and yet others where there does not seem to be a difference (spect). It should be easy to make a more principled assessment of the relation between ratio and AUC, e.g., by computing some sort of correlation coefficient.

The paper is rather well written but would require a thorough proof-read (some examples below, but they are not exhaustive). I think that section 2 can be improved, which is currently a mix of an introduction to Boolean kernels and related work. I think that this section should be devoted to Boolean kernels, and provide a somewhat more detailed introduction to them. For example (line 74), I don't think you can assume that the reader knows what a "monotone" DNF is. I also did not understand why the domain of x and y is R^n and not {0,1}^n. Also the notion of "symmetry" could be explained already here (the description in section 3 is a bit terse). Also expand a bit more your own prior work (briefly summarized in lines 100-106). Maybe section 3 could then also be subsection of the new section on Boolean kernels.

Other works that do not directly tackle Boolean kernels (maybe lines 83-99), could be either a subsection of this section on Boolean kernels, or a section on related work, which I would place at the end (before the conclusions).

Other than that, I only have a few minor comments for improving the paper (numbers refer to line numbers):

  • 2: "Since the down of AI...": I can imagine that you mean "dawn" here. :-)
  • 19: I think we understand quite well what happens "under the hood" of deep neural networks, we just cannot interpret the results.
  • throughout the paper: There is no capitalization in English, excetp for proper names. It is therefore "Boolean kernel" (capitalized, because George Boole was a person) but "propositional kernel" (not capitalized, because "proposition" is not a proper name). Similarly, you should should also not capitalize "support vector machines", "neural networks", or "decision trees", etc.
  • 75: The Sadohara's DNF kernel -> remove "The"
  • 129: considerations cannot be "done".
  • Also in this paragraph: I think the mD kernels and mC kernes have not been introduced before (this should best happen in an expanded section 2, as noted above).
  • 133: it contain -> it contains
  • 136: is the formula '(a) \lor b' correct? If so, I think the parantheses should be explained.
  • 155: non indent here.
  • 165: as much general as possible -> as general as possible
  • 167: we leverage on a -> we leverage a
  • 167: Actually, what you build on here is not a "result". This is a trivial grammar for operators, which is immediately obvious to everybody familiar with grammars (and would not become any clearer to those who are not). I am not even sure whether you would need a citation or a definition here (but I suggest to keep the citation).
  • Example 1: In the second line, I think there is something wrong with the formula.
  • Table 3: The table contains 10 entries, and you write that 4 are missing (including those for true and false). These are 14 in totals. 2 more missing for the full set of 16?
  • 227: b \rightarrow b should be b \rightarrow a, I think.
  • 294: is an hard -> is a hard. And what is a "hard SVM" anyways?
  • Please also carefully check your references, provide complete citations (page numbers), full conference names, consistent formatting. E.g., reference 2 went completely havoc.

Reviewer 2 Report

The authors extend the previous work of Polato (A Novel Boolean Kernels Family for Categorical Data) by enhancing limitations of a) not repeatability use of clauses by Boolean kernel, b) symmetrical consideration of feature in feature space, and c) rigidity of operation which do not allow to use different degree of operation on clauses. The authors explain systematically how they became inspired by the previous work, what are the limitations of previous work and how it can be enhanced by introducing “propositional kernels”. By showing different examples and comparing the earlier work and current proposal they prove that the proposal is logical and how it can be beneficial. They show the basic definitions of their propositional kernels which later they use them for building blocks to construct kernels for any propositional formula.  

I think they are successful to show their point. However, I would like to point out some comments:

  • The author mention: “The only exceptions where the logic equivalences do not hold for the propositional kernels is when there are constraints in the variables.”. The authors explain what it means but they do not explain how this can affect the implementation of their suggestion (i.e. how often we have constraints in the variables, is it often or is it seldom?)
  • In their artificial experiment they used fixed values, why did they come with these values?
  • In algorithm 2, the h is explained as decay (no more explanation in the algorithm) and then in the text they mention:” by changing h is possible to change the degrees range. High values of h decrease the range, while smaller values (h -> 0) would allow to have rules with hundreds of literals.”. Maybe it would be better if they could explain more in detail how h (which is a decay factor) has impact on the degrees of range.
  • They mention: “A drawback of these types of kernels (i.e., not the suggested kernel) is the exponential growth of the size of the feature space. Despite it is possible to reduce the negativity effect of growth by considering kernel of certain degree (e.g., p), the authors suggestion (see Table 3) means the |K| has become square (nA nB) for most of operations (i.e., at most becomes square of p). Maybe it will be clearer if they could elaborate this issue.
  • For real classification problems, the authors mention two criteria to select appropriate prepositional kernels. Then they use algorithm 2 and them minimization method (radius-margin ratio). The question here why algorithms and such minimization? To me it seems they have got a lot of options to how generate the kernels (i.e., this would be naturally due to generalization of their method). Then the algorithm 2 is used (with its predefined value) to decrease the generality. This is paradoxical (i.e., as they write: “We should not know beforehand if and which rule over the input variables produces the labels.”). Maybe the authors had rush to conclude the results but certainly they need to explain it better (i.e., it is not enough to write:” we can enforce a bias toward formulas of a certain length, i.e., shorter rules for ensuring good explainability”). One can say if we can force the outcome why do we need the general rule?
  • It is not explained why the results of datasets of primary-tumore and spect (which even is linear) are less successful?

Reviewer 3 Report

The paper propose a new propositional kernel. The kernel has more expressiveness than boolean kernels based on the composition of other clauses. Overall, I found the paper interesting, but I have some remarks:

  • The claims on line 57-59 and 330-331 are not necessarily true, as you provide an heuristic rule that neither select the kernel which perform best nor have theoretical guarantees of performance. 
  • In lines 323-324, it is not clear how do you choose de C parameter of soft-SVMs or how do you normalize the kernels
  • To me, the statement in lines 161-162 that it is easy to see that you can derive any kernel is not clear. You have provided the example of the xor kernel, that you can't generate.
  • Section 3 is hard to follow. I did not see a connection between non-repeated variables and binomial coefficient (lines 117-117), and I am not sure that "symmetric" is the right word for what you describe from the form of Binary kernel. Finally, I did not understand the last paragraph in this section (lines 138-143).
  • What does the color means in the bullets in Figure 5? 
  • There are a few typos. Please revise the English carefully.

Round 2

Reviewer 3 Report

Authors have addressed my main concerns, and I have no further comments.